

# Annealed averages in spin and matrix models

**Laura Foini**[1][⋆] **and Jorge Kurchan**[2]

**1** IPhT, CNRS, CEA, Université Paris Saclay, 91191 Gif-sur-Yvette, France
**2** Laboratoire de Physique de l'ENS, Ecole Normale Supérieure,
PSL Research University, Université Paris Diderot, Sorbonne Paris Cité,
Sorbonne Universités, UPMC Univ. Paris 06, CNRS, 75005 Paris, France

⋆ laura.foini@ipht.fr

## Abstract

A disordered system is denominated 'annealed' when the interactions themselves may evolve and adjust their values to lower the free energy. The opposite ('quenched') situation when disorder is fixed, is the one relevant for physical spin-glasses, and has received vastly more attention. Other problems however are more natural in the annealed situation: in this work we discuss examples where annealed averages are interesting, in the context of matrix models. We first discuss how in practice, when system and disorder adapt together, annealed systems develop 'planted' solutions spontaneously, as the ones found in the study of inference problems. In the second part, we study the probability distribution of elements of a matrix derived from a rotationally invariant (not necessarily Gaussian) ensemble, a problem that maps into the annealed average of a spin glass model.



# 1 Introduction

Consider the following problem: we are given a system of size $N$ depending on disorder variables $J$, and a set of distributions $P_J$, that we expect to depend exponentially in $N$. The most usual example is a thermodynamic system of, say, $N$ spins with random interactions $J_{ij}$. The corresponding distribution of energies of configurations $e^{S_J(E)} \propto P_J(E)$ defines the (disorder-dependent) entropy. We now consider many samples, and wish to average them: we may choose a *quenched* average $\langle \ln P_J(E) \rangle_J$ or an *annealed* average $\ln \langle P_J(E) \rangle_J$. The two results are generally different [1,2]. In physical, finite-dimensional systems with short-range interactions, one can imagine the system as being composed of many quasi-independent parts, and conclude that the free-energy, energy and entropy (but not their exponentials) are the addition of their values in these parts. This argument suggests that these are the quantities that have to be averaged, since they are the ones that concentrate in probability in the observed value, their average that gives the typical results for one sample.

Annealed averages apply when the disorder evolves in equilibrium with the spins, so there is no reason that they should be treated differently. They have been less studied, but there are reasons to try to understand them better. The origin of our motivation is the following problem: we are given a large $N \times N$ matrix $A$, drawn from a rotationally invariant (generically non-Gaussian) distribution $\mathcal{P}(A) = \mathcal{P}(UAU^\dagger)$, where $U$ may be orthogonal, unitary or symplectic [3,4]. A well-studied example is a 'matrix model' $\mathcal{P}(A) \sim e^{-N\mathrm{tr}W(A)}$, for some potential $W$ [5]. We wish, for example, to know what is the probability marginal distribution of the diagonal $P_{A_{ii}}$ (or equivalently $P_{\sigma \cdot A\sigma}$ with $\sigma$ a $N$-component vector), or of the off-diagonal element $P_{A_{ij}}$. Now, as it is well-known, the integration over all random matrices $A$ may be split in an integration over eigenvalues and another over 'angle' variables defining the eigenbasis. As we shall see, making a quenched calculation amounts to treating eigenvalues of the disorder matrix as fixed and eigenvectors as annealed, although they are originally variables on an equal footing describing the matrix $A$. In conclusion, this is an instance in which the annealed calculation seems the natural one. As we shall see, the result of both approaches differ, and this shows up in the large deviations of the probability distributions.

Our study of the annealed average of spin-glass models shows that the freedom of the couplings to adapt to the spin configurations leads at low temperatures to self-planted solutions [6]. These are configurations with particular low energy with respect to the quenched ones as a result of the annealing. The understanding of this phenomenon that we review and discuss in detail in the first part of our work, is somehow scattered in the literature. In more detail, Hidden Mattis phases, a particular instance of this, have indeed been discussed long ago (see section 2) in [7,8] and the idea of self-planting due to time evolving disorder has been pointed out more recently in the perceptron model [9]. Slowly varying interactions with applications to physics and biology have also been considered in [10,11]. There is also a closely

related computation of Dean and Majumdar [12,13], which involves the large deviations of the lowest eigenvalue of a Gaussian matrix: here we are interested on those towards lower values, the one towards higher values is not relevant here. Yet another example that is easy to understand is the high-pressure phase of spheres with polydispersity. If polydispersity is left to vary freely, each particle will expand as much as allowed, giving rise to a packing with very different statistical properties (see [14,15]).

In the first part of this work (section 2) we discuss in general the global picture which emerges when one considers annealed averages in statistical models with disorder. In the second part (section 3) we proceed to calculate the (annealed) joint distribution of an $r \times r$ submatrix ($r$ finite) of a large $N \times N$ random matrix derived from a general rotationally invariant matrix model.

## 2 Annealed averages in statistical models

In this part of the paper we address the question of what happens to a model when we allow its disorder to adapt, i.e. when we perform an annealed average. Some of the main features may be seen more clearly in the simple case of spherical models, which we shall review first.

### 2.1 Spherical SK model

Working with continuous spins, we have at our disposal the possibility of solving the problem diagonalizing the interaction matrix, so that the discussion is particularly simple. The partition function reads:

$$Z_J(\beta) = \int d\mathbf{s} \, e^{\frac{1}{2}\beta \sum_{ij} J_{ij} s_i s_j} \delta\left(\sum_i s_i^2 - N\right) = \int d\tilde{\mathbf{s}} \, e^{\frac{1}{2}\beta \sum_k \lambda_k \tilde{s}_k^2} \delta\left(\sum_i \tilde{s}_i^2 - N\right), \qquad (1)$$

with $d\mathbf{s} = \prod_i^N ds_i$ and where in the second step we have diagonalized the Hamiltonian. We consider here the case of $J$ real and symmetric. For a rotationally invariant orthogonal ensemble with potential $\mathcal{P}(J) \sim e^{-\frac{N}{2} \text{tr} V(J)}$ the eigenvalues $\lambda_i$ are distributed according to the energy $N^2 E(\boldsymbol{\lambda})$ with:

$$E(\boldsymbol{\lambda}) = \frac{1}{2N} \sum_i V(\lambda_i) - \frac{1}{N^2} \sum_{i>j} \log|\lambda_i - \lambda_j|. \qquad (2)$$

Note that both terms in the action $E(\boldsymbol{\lambda})$ are of order one and one can hope for a non trivial large $N$ limit. As mentioned in the Introduction, there are two possibilities to perform the average over the coupling matrix $J$ (or its eigenvalues). From the statistical physics point of view, one is usually interested in the quenched average of the partition function. Here we do both in preparation for the second part of the paper dedicated to matrix models, where annealed averages are more relevant (for discussion of quenched versus annealed averages in random matrix ensembles see also Ref [16]).

- *Quenched average.* We first draw the $\lambda_i$ from the probability distribution $P(\boldsymbol{\lambda}) \propto e^{-N^2 E(\boldsymbol{\lambda})}$. For example, in a Gaussian ensemble this leads to the semicircle distribution for the $\lambda_i$, for large $N$ [3]. Then, at fixed $\lambda$, we compute expectations of functions of the $s_i$ based on (1) [17]. In practice, this procedure leads to the following average

$$\langle \ln Z_J(\beta) \rangle = \int d\boldsymbol{\lambda} e^{-N^2 E(\boldsymbol{\lambda})} \ln\left\{ \int d\tilde{\mathbf{s}} \, e^{\frac{1}{2}\beta \sum_k \lambda_k \tilde{s}_k \tilde{s}_k} \, \delta\left(\sum_i \tilde{s}_k^2 - N\right) \right\}. \qquad (3)$$

Note that the average over the spins $\tilde{s}_i$, and consequently on the eigenvectors, is annealed. The logic is simple: the distribution of the spins, however peaked, will not

distort the measure of the eigenvalues in the large $N$-limit, because it is under a logarithm.

- *Annealed average.* We treat the $\lambda_i$ and the $\tilde{s}_i$ on an equal footing, so that we consider the combined measure:

$$\langle Z_J(\beta) \rangle = \int d\boldsymbol{\lambda} d\tilde{\mathbf{s}} \, dz \, e^{-N^2 E(\boldsymbol{\lambda}) + \frac{1}{2}\beta \sum_k \lambda_k \tilde{s}_k \tilde{s}_k} \, \delta\left(\sum_i \tilde{s}_k^2 - N\right). \tag{4}$$

## 2.2 The electrostatic analogy

Introducing a Lagrange multiplier in (1) we obtain:

$$Z_J(\beta) = \int d\tilde{\mathbf{s}} \, dz \, e^{\frac{1}{2}\beta \sum_k \lambda_k \tilde{s}_k \tilde{s}_k - \frac{z}{2}\sum_k \tilde{s}_k^2} e^{\frac{Nz}{2}} = \int dz \, e^{\frac{N}{2}\left[z + \log 2\pi - \frac{1}{N}\sum_k \log(z - \beta\lambda_k)\right]}. \tag{5}$$

In the following we will make the change of variable $z \to \beta z$.

For the *quenched* solution we have to solve first for the $\lambda_i$:

$$V'(\lambda_i) - \frac{2}{N}\sum_{j(\neq i)} \frac{1}{\lambda_i - \lambda_j} = 0, \tag{6}$$

a classical exercise in random matrix theory. Then, at fixed $\lambda_i$ we need to solve the dispersion equation:

$$\beta = \frac{1}{N}\sum_k \frac{1}{z - \lambda_k} \tag{7}$$

to determine $z$. We may think (7) as an electrostatic equation in one dimension. We need to find the value of $z$ where the 'electric field' reaches the value $\beta = \frac{1}{T}$. At small values of $T$, the solution is a continuous branch. Depending on the distribution of eigenvalues, in particular for a semicircle distribution, the 'electric field' reaches the vicinity of the rightmost pole with a finite value, and grows sharply at distance $O(1/N)$ of the last pole. This situation is depicted in Figure 1.

From the point of view of large-$N$, this is often described by saying that the root 'sticks' at the value of the largest pole. Note that this situation of finite field up to very close to a system of charges, and then divergence close to the charges, is the usual one in a charged metal! If the 'charge density' instead falls at the edge sufficiently fast, then the 'electric field' does not reach a finite limit at distances $O(1)$ from the edge and there is no freezing mechanism, and hence no transition. An example of this is a constant density of eigenvalues.

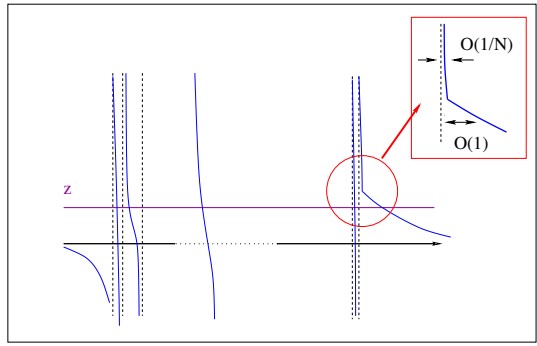

Figure 1: The dispersion relation and the mechanism of freezing: in the large $N$ limit the curve has, to the right of the last pole, a jump in the derivative.

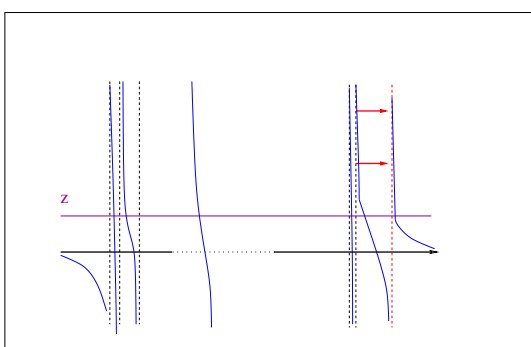

Figure 2: Annealed case: the transition is avoided by detaching an eigenvalue, which stays close $(O(1/N))$ to the value of $z$.

For the *annealed* solution we need to minimize:

$$\frac{1}{2}\sum_i V(\lambda_i) - \frac{1}{N}\sum_{i>j}\log|\lambda_i - \lambda_j| - \frac{\beta}{2}z + \frac{1}{2N}\sum_k \log(z - \lambda_k) + \text{const}. \tag{8}$$

We have now $N$ negative charges and one positive charge at $(\lambda_i, z)$ which have to be treated on an equal footing. Note however that $V'$ acts only on the $\lambda_i$ charges, while there is a linear potential only on $z$. For large temperatures the solution is the same as in (6), because then the charge at $z$ is far and, being a single one, has negligible effect on the bulk. When $T$ is such that $z$ approaches the last charge of the bulk, something remarkable happens: positive and negative charges $(z, \lambda_N)$ form a "molecule" of "size" $|z - \lambda_N| = O(1/N)$ which breaks loose from the bulk and moves to the right. The molecule is subject to interaction with $V$ via $\lambda_N$, and with the linear potential via $z$. The solution where these forces cancel is depicted in Fig. 2.

From the point of view of the interactions $J$ one has a detached eigenvalue and therefore the coupling matrix can be decomposed in $\sum J_{ij}s_i s_j = \sum J'_{ij}s_i s_j + \lambda_N(\sum_i v_i s_i)^2$, where $J'$ is a matrix very similar to the unperturbed one, and **v** is the normalized eigenvector with the detached eigenvalue $\lambda_N$. This last term constitutes a ferromagnetic (or rather 'Mattis') term [18]. This is the mechanism developed by the interaction to lower the free energy. Note also that this form of the coupling matrix is the one used in the so-called planted ensemble for the study of inference problems, where the vector $v$ represents the signal that one aims to recover [6]. In this context in fact, it is known that a rank one perturbation can shift the largest eigenvalue of the original matrix [19].

## 2.3 The solution in terms of the R-transform

Here we describe a calculation that shows explicitly how the mechanism of detaching an eigenvalue occurs for the spherical model and that the annealed free entropy (4) is analytic at all temperatures. We will make use of the Stieltjes transform which is defined as

$$S(z) = \frac{1}{N}\sum_{k=1}^N \frac{1}{z - \lambda_k} = \int \frac{\rho(\lambda)}{z - \lambda}\mathrm{d}\lambda, \tag{9}$$

in the limit of large $N$ and with $\rho(\lambda)$ the (averaged) asymptotic eigenvalue distribution of the matrix $J$. The R-transform is defined from the inverse of the Stieltjes

$$R(\omega) = S^{-1}(\omega) - \frac{1}{\omega}. \tag{10}$$

The R-transform admits a representation in terms of a series expansion where the coefficients can be determined explicitly and are called free cumulants [20]:

$$R(\omega) = \sum_{k=1}^{\infty} C_k \omega^{k-1}. \tag{11}$$

From (10) the R-transform is defined on the real axes in $\omega \in S([\lambda_-, \lambda_+]^c)$, with $\lambda_\pm$ the boundary of the support of the density of eigenvalues, but one can consider its analytical continuation as a complex function [17,20]. Its radius of convergence in (11) can be larger and can be continued beyond $\omega = S(\lambda_+)$.

Let us first see the result at high temperatures for the quenched calculation (3). In terms of these transforms Eq. (7) reads $z = S^{-1}(\beta) = R(\beta) + \beta^{-1}$. This solution is valid when $z \geq \lambda_+$ namely $S(\lambda_+) \geq \beta$. For high temperature one can therefore conclude that the quantity $\Phi = \frac{2}{N} \log Z$ is given by [21–24]:

$$\Phi \sim \int_0^\beta R_\rho(x) \mathrm{d}x. \tag{12}$$

As we discussed above, at $\beta_c$ there is a phase transition where the Lagrange multiplier $z$ sticks on the boundary of the spectrum [17] and consequently the free energy has a non-analiticity.

In the annealed computation one minimizes at the same time over $z$ and over the eigenvalues the function (8). Assuming as above that at low temperatures the eigenvalue $\lambda_N$ separates from the bulk and $z = \lambda_N + \frac{1}{N}a(\lambda_N)$, the eigenvalues in the bulk satisfy the following equations [4,5]:

$$V'(\lambda_i) = 2 \, p.v. \int \mathrm{d}\lambda \frac{\rho(\lambda)}{\lambda_i - \lambda} + O(1/N), \tag{13}$$

for $i = 1, \dots, N-1$, where $p.v.$ stands for principal value, while for the external eigenvalue $\lambda_N$:

$$V'(\lambda_N) = 2S(\lambda_N) + \frac{1}{a(\lambda_N)}. \tag{14}$$

In order to make the mechanism of the annealed solution explicit, we restrict here to the case in which the potential is Gaussian, namely $V(x) = x^2/2$ because in that case it is possible to get explicit expressions for $S$ and $R$ but we believe it applies to a much wider class of rotationally invariant ensembles, as most of our derivation is quite general. For a matrix ensemble with a compact support the Stieltjes transform takes the form [5,24]:

$$S_\pm(x) = \frac{1}{2}V'(x) \pm Q(x)\sqrt{(x - \lambda_+)(x - \lambda_-)}, \tag{15}$$

where $Q(x)$ is a polynomial and $\lambda_\pm$ are the boundaries of the support of the density of eigenvalues. The sign in front of the square root in (15) is chosen such that $S(x) \to 1/x$ for $|x| \to \infty$. Assuming that the minus sign describes the solution at large positive $x$, this implies that

$$\frac{1}{a(\lambda_N)} = 2Q(\lambda_N)\sqrt{(\lambda_N - \lambda_+)(\lambda_N - \lambda_-)}, \tag{16}$$

which also implies that one can read the solution on the "non-physical" branch of the Stieltjes transform because:

$$S_-(\lambda_N) + \frac{1}{a(\lambda_N)} = S_+(\lambda_N), \tag{17}$$

where $S_- = S$ at the right of the support of the spectral density. In fact this is the equation satisfied by the Lagrange multiplier $z = \lambda_N + \frac{1}{N}a(\lambda_N)$:

$$\beta = S_-(\lambda_N) + \frac{1}{a(\lambda_N)} = S_+(\lambda_N), \tag{18}$$

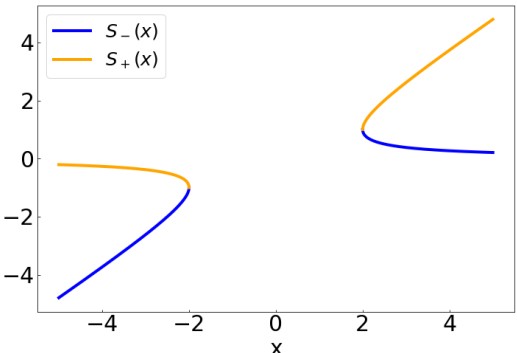

Figure 3: Stieltjes transform for the Gaussian ensemble $S_\pm(x) = \frac{1}{2}(x \pm \sqrt{x^2 - 4})$. The density of eigenvalues is defined between $[-2, 2]$. The physical solution is the one that goes to zero as $1/x$ in the limit $|x| \to \infty$. However its continuation on the other side of the support of the density of eigenvalues intervenes in the solution of the annealed problem.

which implies that for the annealed calculation the solution (12) holds for all temperatures because $S_+^{-1}$ is the continuation of $S^{-1}$ at larger values of $\beta$.

The point of this calculation is that the complex mechanism of detachment of a single eigenvalue and formation of a 'molecule' reduces in this formalism to just following the 'unphysical' branch of the Stieltjes transform, as shown in Figure 3. Understanding this better in the most general non-Gaussian case deserves a deeper analysis.

## 2.4 Two energies and two (un)constrained replicas

In this section we consider the annealed solution of the same problem, with two sets of spins and two energies. This turns out to be important for the discussion of matrix elements in random matrix models in the second part of the work. At this point it is important to specify whether the two replicas are free to overlap, or they are forced to be orthogonal. The latter will be the case for the computation in matrix models Section 3.3. The presence of this constraint leads to two different solutions.

We consider the following annealed average:

$$\langle Z_J(\beta_1, \beta_2) \rangle = \int dJ \, P(J) \int d\mathbf{s}^1 \, d\mathbf{s}^2 e^{\beta_1 \sum_{i<j} J_{ij} s_i^1 s_j^1 + \beta_2 \sum_{i<j} J_{ij} s_i^2 s_j^2} \, \delta\left(\sum_i s_i^1 - N\right) \delta\left(\sum_i s_i^2 - N\right). \quad (19)$$

In the case where the two replicas are constrained to be orthogonal at low enough temperature two eigenvalues detach from the bulk. If this constraint is not imposed it leads to a different solution. To see this we introduce Lagrange multipliers and we proceed in a similar way as before:

$$\langle Z_J(\beta_1, \beta_2) \rangle = \int d\boldsymbol{\lambda} e^{-N^2 E(\lambda)} \int d\tilde{\mathbf{s}}^1 \int d\tilde{\mathbf{s}}^2 e^{\frac{1}{2}\beta_1 \sum_k \lambda_k (\tilde{s}_k^1)^2 + \frac{1}{2}\beta_2 \sum_k \lambda_k (\tilde{s}_k^2)^2 - \frac{1}{2}\beta_1 z_1 (\sum_k \tilde{s}_k^1 - N) - \frac{1}{2}\beta_2 z_2 (\sum_k \tilde{s}_k^2 - N)}$$

$$\propto \int d\boldsymbol{\lambda} e^{-N^2 E(\lambda)} e^{\frac{N}{2}\left[-\frac{1}{N}\sum_k \log(z_1 - \lambda_k) - \frac{1}{N}\sum_k \log(z_2 - \lambda_k) + \beta_1 z_1 + \beta_2 z_2\right]}.$$

$$(20)$$

Our electrostatic problem has now $N + 2$ charges. We need to minimize:

$$\frac{1}{2}\sum_i V(\lambda_i) - \frac{1}{N}\sum_{i>j} \log|\lambda_i - \lambda_j| - \frac{1}{2}(\beta_1 z_1 + \beta_2 z_2) + \frac{1}{2N}\sum_k \log(z_1 - \lambda_k) + \frac{1}{2N}\sum_k \log(z_2 - \lambda_k). \quad (21)$$

The saddle point over $z_1$ and $z_2$ gives:

$$\beta_i = \frac{1}{N} \sum_k \frac{1}{z_i - \lambda_k} \,. \tag{22}$$

The solution at high temperatures $\beta_1$ and $\beta_2$ is as two independent replicas with the unperturbed set of eigenvalues for $J$. For low enough temperatures, *both positive*, one eigenvalue detaches for the bulk and both $z_i$ multipliers attach to it, forming a three-component molecule $(\lambda_N, z_1, z_2)$ of size $1/N$. For temperatures close to zero *but of opposite sign*, there is one eigenvalue at each extreme of the bulk that detaches: the system has a "molecule" to the right $(z_1, \lambda_N)$, and another $(z_2, \lambda_1)$ to the left of the bulk. The situation with opposite temperatures is relevant for the computation of off-diagonal matrix elements in matrix models (see Section 3.4).

## 2.5 The solution of two unconstrained replicas in terms of the R-transform

Let us see in detail the case of positive, low temperatures. At high temperature the eigenvalue distribution is not modified and the result of the partition function is that of two independent replicas:

$$\langle Z_J(\beta_1, \beta_2) \rangle \propto e^{\frac{N}{2} \left( \int_0^{\beta_1} R(x) \mathrm{d}x + \int_0^{\beta_2} R(x) \mathrm{d}x \right)} \,. \tag{23}$$

We first assume now that both $\beta_1$ and $\beta_2$ are large and both Lagrange multipliers are close to the detached eigenvalue $\lambda_N$, namely $z_1 = \lambda_N + a_1/N$ and $z_2 = \lambda_N + a_2/N$. This ansatz translates into:

$$\beta_i = S(\lambda_N) + \frac{1}{a_i} \,, \tag{24}$$

and the equation for the largest eigenvalue reads:

$$V'(\lambda_N) = 2S(\lambda_N) + \frac{1}{a_1} + \frac{1}{a_2} \,, \tag{25}$$

which combined with (24) gives:

$$\beta_1 + \beta_2 = V'(\lambda_N) \,. \tag{26}$$

Assuming now that one eigenvalue $\lambda_N$ has detached and that only one Lagrange multiplier is close to it, which occurs when $\beta_1 > S(\lambda_N)$ and $\beta_2 < S(\lambda_N)$ (or the same with exchange of $\beta_1$ and $\beta_2$), the second Lagrange multipliers sticks to it under the condition:

$$\beta_1 = S_+(\lambda_N) \qquad \beta_2 = S(\lambda_N) \,. \tag{27}$$

In particular for the Gaussian ensemble this implies $\beta_1 \beta_2 = 1$. Note that at this transition point the condition of analyticity is not ensured, as it was also found in [25].

Let us now see the case of opposite temperatures $\beta_1 > 0$ and $\beta_2 < 0$. At high temperature the partition function is again (23) and the spectrum of the eigenvalues is unperturbed. However one has to be careful with the continuation of this result to small temperatures. In this limit two eigenvalues detaches from the boundary left and right when

$$\beta_1 = S(\lambda_N) \qquad \text{and} \qquad \beta_2 = S(\lambda_1) \,. \tag{28}$$

We suppose as before that $z_1 = \lambda_N + \frac{1}{N} a(\lambda_N)$ and $z_2 = \lambda_1 - \frac{1}{N} b(\lambda_1)$. The equations for the detached eigenvalues $\lambda_1$ and $\lambda_N$ read

$$V'(\lambda_N) = 2S_-(\lambda_N) + \frac{1}{a(\lambda_N)} \,, \tag{29}$$

$$V'(\lambda_1) = 2S_+(\lambda_1) - \frac{1}{b(\lambda_1)}, \tag{30}$$

and the equation for $z_1$ and $z_2$ become:

$$\beta_1 = S_-(\lambda_N) + \frac{1}{a(\lambda_N)} = S_+(\lambda_N), \tag{31}$$

$$\beta_2 = S_+(\lambda_1) - \frac{1}{b(\lambda_1)} = S_-(\lambda_1), \tag{32}$$

therefore they are both read from the analytic continuation at low temperatures of the inverse of the Stieltjes and the solution (23) holds at all temperatures.

## 2.6 Low temperature of the annealed model and large deviations

In this section we make a digression to emphasize that two models may have essentially the same quenched properties, but very different low-temperature annealed behavior. This may come as a surprise, given that the latter is just the analytic continuation of the high-temperature phase: the continuations of two almost identical high temperature behaviors become widely different at sufficiently low temperatures.

Consider a Gaussian $V_G(x) = x^2/2$ matrix model, and a slightly perturbed one $V_{NG}(x) = x^2/2 + g/4x^4$. The Stieltjes transforms are $S_G(x) = \frac{1}{2}(x - \sqrt{x^2 - 4})$ and $S_{NG}(x) = \frac{1}{2}(x + gx^3 - (1 + gx^2 + 2ga)\sqrt{x^2 - 4a^2})$ respectively [5] (here $a$ is the solution of $3ga^4 + a^2 - 1 = 0$). For small $g$, $S_G$ and $S_{NG}$ do not differ very much at any $x$. The density of eigenvalues is also almost the same. If we now move to the low temperature annealed problem, one eigenvalue detaches, and enters the region of large $x$, where the behavior $V'_G \propto x$ for the Gaussian and $V'_{NG} \propto x^3$ for the perturbed one play an important role, determining the behavior of $S_{G+}$ and $S_{NG+}$, the expressions of $S_G$ and $S_{NG}$ with a positive sign in front of the squared root, considered for $x \gg 1$. This means that the two models while behave very similarly in the high temperature phase (their energy scales as $E_G^{T \gg 1} \sim E_{NG}^{T \gg 1} \sim -\beta = -1/T$), they are very different at low temperature, where the Gaussian model has an energy that goes as $E_G^{T \ll 1}(T) \sim -1/T$ while for the quartic potential $E_{NG}^{T \ll 1}(T) \sim -T^{-1/3}$. As we shall see below, this effect manifests in the probability tails of a diagonal matrix element drawn from a distribution with a potential $V_{NG}$, which scales as: $\ln P(A_{ii}) \propto -NA_{ii}^4$, for large $A_{ii}$.

## 2.7 Complex annealed landscape: the case $p > 2$

Let us now generalize the previous results to a problem with a more complex landscape. The simplest of them all is a spherical model with $p$-body interactions $p > 2$:

$$E_J^p[\mathbf{s}] = -\sum_{i_1 \dots i_p} J_{i_1 \dots i_p} s_{i_1} \dots s_{i_p} \quad ; \quad \sum_i s_i^2 = N. \tag{33}$$

For Gaussian couplings the model is well-studied. The ensemble of $J_{i_1 \dots i_p}$ is rotationally invariant in this case, it is a tensor generalization of a Gaussian matrix model. Non-Gaussian variants are also possible, although we shall not study them here. From now on we concentrate on the case $p = 3$.

The quenched solution has a phase transition while the annealed one

$$\langle Z_J^p(\beta) \rangle = \int \mathrm{d}J P(J) \int \mathrm{d}\mathbf{s}\, e^{-\beta E_J^p[\mathbf{s}]} \delta\left(\sum_i s_i^2 - N\right) \tag{34}$$

(superficially) does not [26].

The notion of 'detaching an eigenvalue' may be expected to generalize as the fact that below the (quenched) critical temperature, the annealed model develops a 'spike' in the interaction $J_{ijk} \to J'_{ijk} + \frac{a(T)}{N^2} v_i v_j v_k$, where the $J'$ have the same statistical properties as the $J$ (and are of $O(N^{-1})$), and $a(T)$ and $v_i$ are of order one. We have evidence for this of two sorts: dynamical and static.

*i)* The dynamic case has actually been done for us many years ago [27]: Barrat *et al.* computed the evolution starting from an equilibrated configuration, obtained from an annealed ansatz, as befits a system above the static transition temperature, the Kauzmann (replica symmetry breaking) temperature $T_k$. In the high temperature phase, the dynamics were completely ergodic, while in a regime of intermediate temperatures $T_k < T < T_d$ between the static and the dynamic transitions, it was found that, as expected, the system is confined to one of the many states that together constitute the Gibbs measure in that regime. What is relevant for us here is that if we extend their annealed solution below the static transition of the quenched system, annealed and quenched measures start do differ. The initial annealed state is now a condition that corresponds to biased $J$'s, and the subsequent dynamics allows us to study its characteristics. The dynamics is now inside a deep state whose energy and free-energy have been 'pulled below' the level of the ground state for typical $J$'s. The 'size' of this *self-planted* state, as measured by the Edwards-Anderson parameter of the dynamics, is small. All characteristics of the state are analytic continuations of the equilibrium states contributing to the Gibbs measure in the intermediate temperature phase.

*ii)* Once we know that the state generated by the annealed measure is deep, we still do not know whether it is isolated in phase-space, or a whole cluster of nearby states are 'planted' by the annealed measure. To answer this, there is a static argument, which we shall develop in Section 2.10: it is to show that once we have an annealed $J$ with a deep self-planted ground state, the landscape restricted to configurations orthogonal to that state is the same as the one of the old quenched system. In this sense, the 'spike' is isolated.

## 2.8 The case of disorder on a lattice

Here we consider a generalization of annealed averages to finite dimensional models. Here, in general, the condition of rotational invariance of the coupling matrix $\mathcal{P}(J) = \mathcal{P}(UJU^\dagger)$ for an orthogonal matrix $U$ is not satisfied. When the interactions have a lattice structure that is not destroyed by annealing, there are clearly limitations on the spectra as the ferromagnetic lattice can not be realized with a rank one perturbation. The case of a three-dimensional Edwards-Anderson model $E = -\sum_{n.n.} J_{ij} s_i s_j$ with $J_{ij} = \pm 1$ and $s_i$ Ising spins is very clear. All ground states are configurations of the form $J_{ij} = \sigma_i \sigma_j$, for a given set $\sigma_i = \pm 1$. This is a Mattis system, a ferromagnet in disguise, as one may check by gauge-transforming the spins of the system as $s_i \to \sigma_i s_i$.

## 2.9 Diffusion over annealed solutions

Slow dynamics can arise when the system has two different sets of spins. In this Section we study numerically the problem of Ising spins, instead of the spherical ones, showing that similar results hold also in this case. The optimal solution may need to plant one or two deep valleys: if the optimal number is only one, the solution with two valleys is metastable, but may be long-lived. The partition function that we analyze is then the following:

$$\langle Z_J^p(\beta_1, \beta_2) \rangle = \int dJ P(J) \sum_{\sigma^1, \sigma^2 \in \{-1,1\}^N} e^{\beta_1 \sum_{i_1 \dots i_p} J_{i_1 \dots i_p} \sigma_{i_1}^1 \cdots \sigma_{i_p}^1 + \beta_2 \sum_{i_1 \dots i_p} J_{i_1 \dots i_p} \sigma_{i_1}^2 \cdots \sigma_{i_p}^2}, \quad (35)$$

and for simplicity we take $J_{i_1 \dots i_p}$ Gaussian distributed with average zero and variance

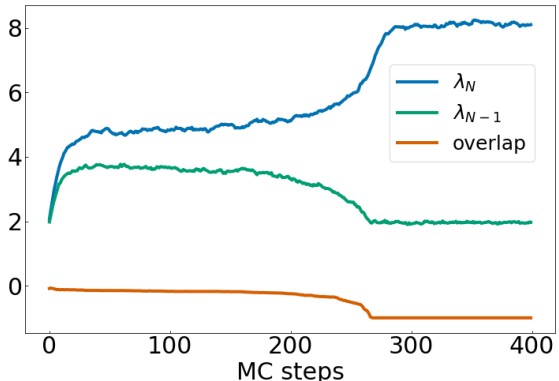

Figure 4: A run with two replicas at temperature $\beta_1 = \beta_2 = 4$ for $p = 2$. We show the first two eigenvalues and the overlap between the two replicas.

$\langle J^2_{i_1 \ldots i_p} \rangle = p!/2N^{p-1}$. The case $p = 2$ corresponds to the replicated annealed SK model. A direct annealed computation of (35) gives the following expression:

$$\langle Z^p_J(\beta_1, \beta_2) \rangle = \int \mathrm{d}q \; e^{N\left[\frac{1}{4}(\beta_1^2 + \beta_2^2 + 2\beta_1\beta_2 q^p) - \frac{1+q}{2}\ln\left(\frac{1+q}{2}\right) - \frac{1-q}{2}\ln\left(\frac{1-q}{2}\right)\right]}, \tag{36}$$

where $q = \sum_i \sigma^1_i \sigma^2_i$ is the overlap between replicas.

The solution of this problem for $p = 2$ predicts a second order phase transition at $\beta_1\beta_2 = 1$ from $q = 0$ to $q \neq 0$. Note however that this solution does not say anything about the spectrum of the matrix $J$ in these two solutions.

In Figure 4 we show a run of a Montecarlo simulation of the model (35) for $p = 2$ with $N = 500$ spins at low temperature ($\beta_1 = \beta_2 = 4$), starting from two uncorrelated configurations. In particular we present the first two eigenvalues: clearly at the beginning of the the simulation two eigenvalues detach form the bulk $\lambda \in [-2, 2]$ signalling a metastable solution with low overlap between the configurations. However after a certain number of sweeps the true equilibrium solution is reached with only one eigenvalue out of the bulk and a perfect overlap between the two configurations (which can be positive or negative with equal probability).

We note that the solution with two eigenvalues out of the bulk is the stable one if one forces the two sets of spins to be orthogonal. In Fig. 5 we show a run in the same condition as before but with an additional coupling $\alpha \left( \sum_i \sigma^1_i \sigma^2_i \right)^2$ with $\alpha \gg 1$ which does this. We see

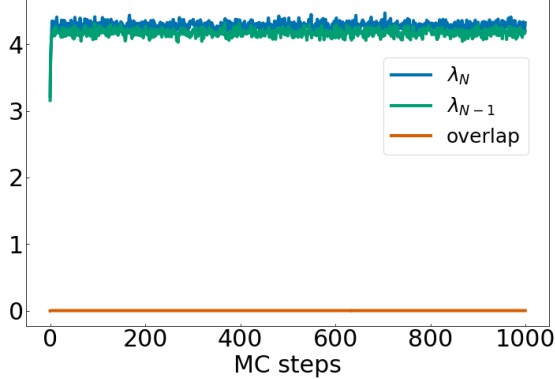

Figure 5: A run with two orthogonal replicas at temperature $\beta_1 = \beta_2 = 4$ for $p = 2$. We show the first two eigenvalues and the overlap between the two replicas.

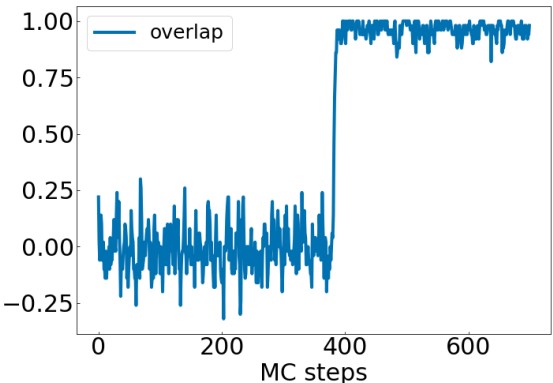

Figure 6: A run with two replicas in the *p*-spin model with $p = 3$ at temperature $\beta_1 = \beta_2 = 1.2$. The overlap between the two replicas stays close to zero until it jumps suddenly to one.

that the solution with two eigenvalues out of the bulk and (obviously) small overlap is the equilibrium stable one in this case.

For $p > 2$ the situation is different. At low temperatures there are two phases separated by a first order phase transition depending on $|\beta_1 \beta_2|$, separating a phase with $q = 0$ where there are no planted states or two orthogonal ones, and one with $q$ near one where both replicas are in the same planted state. In Fig. 6 we monitor the annealed dynamics of the overlap (for $N = 100$ spins) at $\beta_1 = \beta_2 = 1.2$ and we see that it stays close to zero and then suddenly jumps to one, when both copies have 'found one another'.

## 2.10 'Witness' model and order parameter

The free energy (4) is analytic for all $\beta$, and the same is true for all $p$. One expects, however, that some discontinuity shows up at the point in which an eigenvalue suddenly detaches from the distribution, or that a self-planted state appears in the more general situation. When one considers what is happening to the distribution of the $J$'s, the strategy becomes obvious: generate the $J$'s with the annealed process, and use them as the *quenched* disorder of a "witness" model: either by studying its equilibrium, its dynamical properties, or by performing a Kac-Rice study of the saddle points of its potential. An instance of this was already proposed in Ref. [8]. Consider the distribution $\mathcal{P}(J)$ of $J$'s derived from an annealed process (1):

$$\mathcal{P}(J) \propto P(J) \int d\mathbf{s}\, e^{\frac{1}{2}\beta \sum_{ij} J_{ij} s_i s_j} \delta\left(\sum_i s_i^2 - N\right)/\bar{Z}\,, \tag{37}$$

where $\bar{Z}$ is the normalization, and use it as quenched disorder for a system with spins $\sigma_i$:

$$Z(\beta) = \frac{1}{\bar{Z}} \int dJ\, P(J) \int d\mathbf{s}\, e^{\frac{1}{2}\beta \sum_{ij} J_{ij} s_i s_j} \delta\left(\sum_i s_i^2 - N\right) \ln\left\{\int d\boldsymbol{\sigma}\, e^{\frac{1}{2}\beta \sum_{ij} J_{ij} \sigma_i \sigma_j} \delta\left(\sum_i \sigma_i^2 - N\right)\right\}\,. \tag{38}$$

It is clear that a detached eigenvalue in the $J$'s acts as a ferromagnetic (or rather, a Mattis term $(\sum_i v_i \sigma_i)^2$) for the $\sigma_i$, and this will show up in the inter-state correlation. In replica language, Eq (38) may be expressed as:

$$Z(\beta) \propto \langle Z_J(\beta) \log Z_J(\beta) \rangle = \left\langle \frac{\partial}{\partial n} Z_J^n \right\rangle \Big|_{n=1}\,, \tag{39}$$

and the order parameter:

$$q_o = \langle s_a s_b \rangle \quad a \neq b\,, \tag{40}$$

where the problem has $n \to 1$ replicas. Exactly the same strategy may be used for complex landscapes, for example $p > 2$, where there will be a transition to a deep planted state [6]. The strategy may be implemented with two or more families of $\sigma_i$, if more self-planted states are to be detected.

Let us now make the calculation we announced in Section 2.7 for $p = 3$ to show that there are no deep states orthogonal to the planted state. We demand that the spins in the 'witness' model be orthogonal to those in the planted one, and check that with such restriction the model is the same as the quenched one:

$$\tilde{Z}(\beta) = \int dJ P(J) \int ds\, e^{\beta \sum_{i<j<k} J_{ijk} s_i s_j s_k}\, \delta\left(\sum_l s_l^2 - N\right) \times$$
$$\ln\left[\int d\sigma\, e^{\beta \sum_{i<j<k} J_{ijk}\sigma_i\sigma_j\sigma_k}\, \delta\left(\sum_l \sigma_l^2 - N\right)\delta\left(\sum_l \sigma_l s_l\right)\right]. \tag{41}$$

Again, this is problem with $n + 1$ replicas, with $n \to 0$. The constraint that imposes that the $\sigma_i$ be orthogonal to the $s_i$ means, once the replica trick is applied, that the matrix $Q_{ab}$ will have $Q_{1a} = Q_{a1} = 0$ so that the replica matrix breaks into a one by one and an $n \times n$ block. It is a general property of the replica trick that when the matrix breaks into blocks the replicas uncouple. Hence we get the original annealed problem for $Q_{11}$ and an ordinary quenched one for $Q_{ab}$ ($a > 1, b > 1$). Thus, we conclude that the model restricted to be orthogonal to the $s_i$ vector is just as if the $J$ were drawn from quenched ensemble.

# 3 The distribution of elements in matrix models

We wish to compute the joint distribution of a number of elements $\{A_{a_1 b_1}...A_{a_s b_s}\}$ of a matrix model with invariances $\mathcal{P}(A) = \mathcal{P}(UAU^\dagger)$, and we consider here the case where $A$ is real and symmetric and $U$ is an orthogonal matrix. The generalization to the complex case is also possible. The most usual case is when the model is defined through: $\mathcal{P}(A) = e^{-N \operatorname{tr} W(A)}$ (playing the role of the distribution $\mathcal{P}(J) = e^{-\frac{N}{2} \operatorname{tr} V(J)}$ in the previous section). Let $\hat{A}_{ab}$ be an $r \times r$ submatrix, where $a$ and $b$ may be assumed to be $1, ..., r$. In the following we will denote with a symbol ˆ all matrices of small $r = O(1)$ size. In [28] we showed that the probability distribution of a sub-block of size $2 \times 2$ inherits the property of rotational invariance of the original ensemble, namely that

$$P(\hat{A}) = P(\hat{U}\hat{A}\hat{U}^\dagger), \tag{42}$$

where now $\hat{U}$ are $2 \times 2$ elements of the group. Below we will argue that the same proof can be generalized to a block of arbitrary dimension, and we shall relate this calculation of $P(\hat{A})$ to the problem of solving annealed spin-glass models.

## 3.1 Probability distributions and spin-glass models

Let us start, for clarity, with the case $r = 1$. We consider the probability distribution:

$$P_{r=1}(\hat{A}_{11}) = \int dA\, \mathcal{P}(A)\, \delta(\sigma \cdot A\sigma - \hat{A}_{11}), \tag{43}$$

where $\sigma$ is a $N$-component vector with unit norm. Eq. (43) corresponds to the diagonal matrix element of the matrix $A$ in a basis where $\sigma$ in one of the vectors defining that basis. As the ensemble is invariant under change of basis such quantity does not depend on $\sigma$ and we can average over that:

$$P_{r=1}(\hat{A}_{11}) = \mathcal{N}\int dA\, d\sigma\, \mathcal{P}(A)\, \delta(\sigma \cdot A\sigma - N\hat{A}_{11})\delta(\sigma \cdot \sigma - N), \tag{44}$$

where the constant $\mathcal{N}$ ensures the normalization of the probability, and we have rescaled the vector $\boldsymbol{\sigma}$ to have norm $N$ for convenience. The problem of the calculation of one diagonal matrix element has become the calculation of the annealed entropy of a spin-glass, its Laplace transform will be the partition function we studied in the previous section.

Let us now generalize to the probability distribution of a sub matrix of size $r \times r$. We have:

$$P_r(\hat{A}) \;\; = \;\; \mathcal{N} \int \, \mathrm{d}A \, \prod_a \mathrm{d}\boldsymbol{\sigma}_a \, \mathcal{P}(A) \, \delta(\boldsymbol{\sigma}_a \cdot A \boldsymbol{\sigma}_b - N\hat{A}_{ab}) \prod_{ab} \delta(\boldsymbol{\sigma}_a \cdot \boldsymbol{\sigma}_b - N\delta_{ab}), \quad (45)$$

where, again, we used the invariance under change of basis to average over the $\boldsymbol{\sigma}_a$. This is precisely a spin-glass model with $r$ sets of spins, forced to remain orthogonal. The fact mentioned above that $P(\hat{A})$ is invariant with respect to transformations in the $r \times r$ space implies that $P(\hat{A})$ is a function of the eigenvalues of $\hat{A}$. This invariance can be verified directly in Eq (45) by applying a similar reasoning as that described in [28]. In particular one can consider $\hat{U}$ in (45), where $\hat{U}$ acts non trivially on the basis $\boldsymbol{\sigma}_a$. Upon change of variables, and identifying $\hat{U}$ with a $N \times N$ orthogonal matrix that acts as $\hat{U}$ on the space defined by $\boldsymbol{\sigma}_a$ and leaves unperturbed the other $N - r$ vectors of the basis, one indeed checks that $P_r(\hat{U}\hat{A}\hat{U}^\dagger) = P_r(\hat{A})$.

One can also write, for $\mathcal{P}(A) \sim e^{-N\mathrm{tr}W(A)}$:

$$P_r(\hat{A}) = \mathcal{N} \int \, \mathrm{d}A \, e^{-N\mathrm{tr}W(A)} \prod_a \mathrm{d}\boldsymbol{\sigma}_a \, \prod_{ab} d\beta_{ab} dz_{ab} \, e^{\frac{1}{2}\beta_{ab}(\boldsymbol{\sigma}_a \cdot A \boldsymbol{\sigma}_b - N\hat{A}_{ab}) - \frac{1}{2}z_{ab}(\boldsymbol{\sigma}_a \cdot \boldsymbol{\sigma}_b - N\delta_{ab})}, \quad (46)$$

where the integrals over the $\beta_{ab}$ and $z_{ab}$ run over the imaginary axis.

We may now explicitly diagonalize $\beta_{ab} = \sum_c \hat{U}^\beta_{ac} \tilde{\beta}_c \hat{U}^\beta_{bc}$, $z_{ab} = \sum_c \hat{U}^z_{ac} \tilde{z}_c \hat{U}^z_{bc}$ and $\hat{A}_{ab} = \sum_c \hat{U}^A_{ac} \tilde{a}_c \hat{U}^A_{bc}$ and integrate first over the parameters defining $\hat{U}^\beta$ and $\hat{U}^z$. The saddle point over those happens when $z_{ab}$ and $\beta_{ab}$ are both diagonal in the same basis as $\hat{A}_{ab}$. Integrating the $\boldsymbol{\sigma}_a$ away, in terms of the eigenvalues $\lambda_i$ of $A$, the exponent in Eq (46) becomes, distinguishing the $r$ eigenvalues $\tilde{\lambda}_1, ..., \tilde{\lambda}_r$ that may be detached from the bulk:

$$-N \sum_{i \in \mathrm{bulk}} W(\lambda_i) - N \sum_{k=1}^{r} W(\tilde{\lambda}_k) + \sum_{ij \in \mathrm{bulk}} \ln|\lambda_i - \lambda_j| + \sum_{k,k'=1}^{r} \ln|\tilde{\lambda}_k - \tilde{\lambda}'_k| + \sum_{k=1}^{r} \sum_{i \in \mathrm{bulk}} \ln|\lambda_i - \tilde{\lambda}_k|$$
$$-\frac{N}{2} \sum_{k=1}^{r} \sum_{i \in \mathrm{bulk}} \ln(\tilde{z}_k - \lambda_i \tilde{\beta}_k) + \frac{N}{2} \sum_{k=1}^{r} \tilde{z}_k - \frac{N}{2} \sum_{k,k'=1}^{r} \ln(\tilde{z}_k - \tilde{\beta}_k \tilde{\lambda}_{k'}) - \frac{N}{2} \sum_{n=1}^{r} \tilde{\beta}_k \tilde{a}_k. \quad (47)$$

We recognize three kinds of terms:

- a bulk term, just as if the system were quenched, but with (at least) $N - r$ eigenvalues

- a subdominant term of interaction between detached eigenvalues of $O(1)$

- a sum of terms for each detached eigenvalue (and its corresponding $\tilde{z}_n$) with the same form of an $r = 1$ problem formula.

The first contribution gives an $O(1/N)$ correction to the density of eigenvalues of the bulk. The interaction between detached eigenvalues only acts if their differences are $O(1/N)$: it is a correction that couples real replicas. The third contribution is what we are concerned with: it is *superficially* (because these terms potentially interact through their effect on the bulk of $\lambda_i$) a sum of noninteracting terms for each replica, each of the same form of the one of $r = 1$. If we compute the saddle point for each $\tilde{z}_a$, and eliminate then to obtain a form, in terms of the probability of a diagonal term $P^{\mathrm{diag}}(\hat{A}_{11}) = P_{r=1}(\hat{A}_{11})$:

$$\ln P_r(\hat{A}) = \sum_{a=1}^{r} \ln P^{\mathrm{diag}}(\tilde{a}_n) = -\frac{N}{2} \, \mathrm{tr}\,\tilde{W}(\hat{A}), \quad (48)$$

where we have defined $\tilde{W}(x) \equiv -\frac{2}{N} \ln P^{\mathrm{diag}}(x)$. In short: *the large deviation function of an $r \times r$ submatrix $\mathrm{tr}\,\tilde{W}(\hat{A})$ is the same for all finite $r$.*

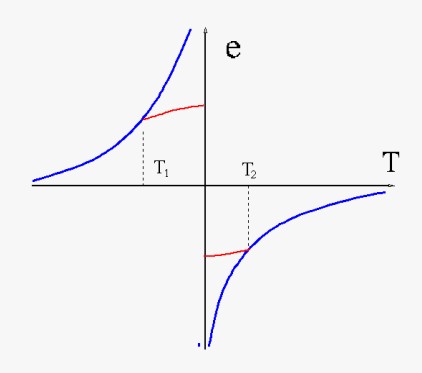
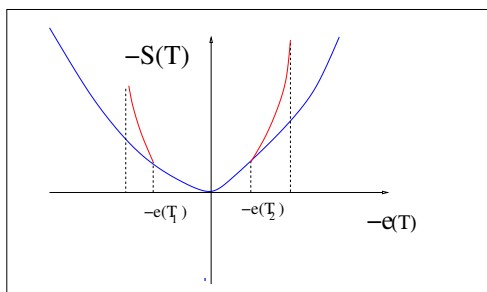

Figure 7: Left: a sketch of an energy versus temperature $(T = \beta^{-1})$ curve for the annealed and quenched problem. Right: the corresponding parametric plot of $-s(T) = -\frac{\partial T\Phi}{\partial T}$ versus $-e(T)$ which gives the large deviation function $\tilde{W}(\hat{A})$ for the probability $P \sim e^{-\frac{N}{2}\tilde{W}(\hat{A})}$ of a diagonal matrix element $\hat{A} = -e$. The range of the quenched probability is bounded, while the one of the annealed problem is not.

## 3.2 Generating functions and marginals

Let us write the Laplace transform of Eq. (46) as

$$\langle e^{\frac{N}{2}\mathrm{tr}\hat{\beta}\hat{A}} \rangle \sim \int \prod_a \mathrm{d}\boldsymbol{\sigma}_a \, dz_{ab} \, dA \, \mathcal{P}(A) \prod_{ab} e^{\frac{1}{2}\hat{\beta}_{ab}\boldsymbol{\sigma}_a \cdot A\boldsymbol{\sigma}_b - \frac{1}{2}z_{ab}(\boldsymbol{\sigma}_a \cdot \boldsymbol{\sigma}_b - N\delta_{ab})} = \langle Z_A(\hat{\beta}) \rangle \sim e^{\frac{N}{2}\Phi(\hat{\beta})} \,, \quad (49)$$

where the first brackets means $\langle \bullet \rangle = \int D\hat{A} \, \bullet \, P_r(\hat{A})$. Eq. (49) is the generating function of Eq. (46). The passage of (45) to (49) is the standard one between microcanonical to canonical, except that there are several 'temperatures'. We recognize the *annealed average over "disorder"* *(=A)* of the spherical model with *r* 'replicas' *that are forced to be orthogonal*, the quantity $\langle Z_A(\hat{\beta}) \rangle$. The thermodynamic relations are (for $N \to \infty$):

$$\hat{A}_{ab} = \frac{\partial \Phi}{\partial \beta_{ab}} \,, \qquad \beta_{ab} = \frac{\partial \, \mathrm{tr}\,\tilde{W}(\hat{A})}{\partial \hat{A}_{ab}} \,. \qquad (50)$$

The quantity $e_{ab} = -\hat{A}_{ab}$ plays the role of energies, the $\beta_{ab}$ of temperatures, and $s(\hat{A}) = \mathrm{Const} - \mathrm{tr}\,\tilde{W}(\hat{A})$ of entropy. Eliminating the temperatures $\beta_{ab}$ in favor of the $\hat{A}_{ab}$ one obtains the desired distribution $\ln P_r(\hat{A})$. This is particularly clear if we think to the case $r = 1$.

The relation between thermodynamics and the probability of diagonal matrix elements is summarized in Figure 7.

If we wish to calculate the marginal distribution of some elements of the $r \times r$ matrix, we need to integrate (46) over some of the $\hat{A}_{ab}$. This, in turn, amounts to setting to zero the corresponding $\beta_{ab}$ in Eq. (49). The remaining variables, those not integrated upon, are given by Eq (50). For example, if we wish to compute the joint distribution of $r$ diagonal elements $\hat{A}_{aa}$, with $a = 1, ..., r$, we set $\beta_{ab} = 0$ for $a \neq b$. Another example is to calculate the probability distribution of an off-diagonal element, we set $r = 2$ and $\beta_{11} = \beta_{22} = 0$, $\beta_{12} = \beta_{21} = \beta$. We shall do this below.

### 3.3 Diagonal matrix elements in matrix models

The marginal joint distribution of $r$ diagonal eigenvalues is obtained by setting $\beta_{ab} = 0$ for $a \neq b$. The analysis is just as above. We thus conclude that, to leading order, the marginal:

$$\ln P(\hat{A}_{11}, ..., \hat{A}_{rr}) = \sum_{a=1}^{r} \ln P^{\mathrm{diag}}(\hat{A}_{aa}),\tag{51}$$

i.e. the diagonal elements are independent for finite $r$. Let us emphasize that this is a large $N$ result, valid for $r$ much smaller than $N$, such that the repulsion between eigenvalues may be neglected.

### 3.4 Off-diagonal matrix elements: two replicas at opposite temperature

Considering $r = 2$ with equal off-diagonal "temperatures" $\beta_{12} = \beta_{21} = \beta$ and zero diagonal terms $\hat{\beta}_{aa} = 0$ in (49) as indicated above, we get:

$$\langle Z_A^{\mathrm{off}}(\beta) \rangle \equiv \int \prod_a \mathrm{d}\boldsymbol{\sigma}_a \prod_{ab} dz_{ab}\, \mathrm{d}A\, \mathcal{P}(A)\, e^{\frac{1}{2}\beta\boldsymbol{\sigma}_1 \cdot A\boldsymbol{\sigma}_2 + \frac{1}{2}\beta\boldsymbol{\sigma}_2 \cdot A\boldsymbol{\sigma}_1} \prod_{ab} e^{-\frac{1}{2}z_{ab}(\boldsymbol{\sigma}_a \cdot \boldsymbol{\sigma}_b - N\delta_{ab})}.\tag{52}$$

The symbol $\langle Z_A^{\mathrm{off}}(\beta) \rangle$ is indicating, as in (49), that such generating function for off-diagonal matrix elements, is equivalent to an annealed average of an associated spin glass problem. In fact, a rotation to $(\boldsymbol{\sigma}_1 \pm \boldsymbol{\sigma}_2)/\sqrt{2}$, and similarly for the $z_{ab}$ leads to:

$$\langle Z_A^{\mathrm{off}}(\beta) \rangle \equiv \int \prod_a \mathrm{d}\boldsymbol{\sigma}_a \prod_{ab} dz_{ab}\, \mathrm{d}A\, \mathcal{P}(A)\, e^{\frac{1}{2}\beta\boldsymbol{\sigma}_1 \cdot A\boldsymbol{\sigma}_1 - \frac{1}{2}\beta\boldsymbol{\sigma}_2 \cdot A\boldsymbol{\sigma}_2} \prod_{ab} e^{-\frac{1}{2}z_{ab}(\boldsymbol{\sigma}_a \cdot \boldsymbol{\sigma}_b - N\delta_{ab})}\tag{53}$$

and therefore one sees that the problem of the generating function of an off-diagonal matrix element maps into two orthogonal replicas with opposite temperatures. Performing the same steps as in the previous section one arrives to the partition function:

$$\langle Z_A^{\mathrm{off}}(\beta) \rangle \propto \int \mathrm{d}z_1 \mathrm{d}z_2 \mathrm{d}z_{12} e^{\frac{N}{2}\left(z_1 + z_2 + \sum_k \log\left[(z_1 - \beta\lambda_k)(z_2 + \beta\lambda_k) - z_{12}^2\right]\right)}.\tag{54}$$

The saddle point over $z_{12}$ which enforces the orthogonality between the two vectors imposes $z_{12} = 0$ and therefore the result is that of two independent diagonal integrals, a particular case of what discussed in Section 2.4 for replicas at opposite temperature (which leads at low temperature to two eigenvalues one to the left and one to the right out of the bulk). The result (23) implies the following relation between diagonal and off-diagonal matrix elements:

$$\langle Z_A^{\mathrm{off}}(\beta) \rangle = \langle Z_A^{\mathrm{diag}}(\beta) \rangle \langle Z_A^{\mathrm{diag}}(-\beta) \rangle,\tag{55}$$

where $\langle Z_A^{\mathrm{diag}}(\beta) \rangle$ is the (annealed) partition function of one single replica at inverse temperature $\beta$, or equivalently $\langle e^{\frac{1}{2}\beta\hat{A}_{11}} \rangle$.

#### 3.4.1 Applications

As an application let's see how these formulas allows us to compute the distribution of matrix elements in some specific ensembles where we know the R-transform. The result (12) allows us to write for the diagonal matrix elements:

$$P_{A_{ii}}(a) \sim e^{\frac{N}{2}\min_\beta\left[-\beta a + \int_0^\beta \mathrm{d}x\, R(x)\right]}.\tag{56}$$

Form this, knowing that $R(x) = x$ of a Gaussian ensemble [20], we get

$$P_{A_{ii}}^G(a) \sim e^{-Na^2/4},.\tag{57}$$

Moreover from (23), (55) and:

$$P_{A_{ij}}(a) \sim e^{N \min_\beta \left[ -\beta a + \frac{1}{2} \int_0^\beta dx\, R(x) + \frac{1}{2} \int_0^{-\beta} dx\, R(x) \right]}, \tag{58}$$

we recover the expected result for the off-diagonal matrix element of a Gaussian ensemble:

$$P_{A_{ij}}^G(a) \sim e^{-Na^2/2}. \tag{59}$$

Let us now analyze a less trivial case. For a Wishart matrix $A = WW^\dagger$ where $W$ is matrix of size $N \times K$ ($N \leq K$) whose entries are Gaussian random variables with zero mean and variance $1/N$, it is known that $R(x) = \alpha/(1-x)$ with $\alpha = K/N$ [20]. With this we obtain the probability distribution of a diagonal matrix element of such matrix, valid in the large $N$ limit:

$$P_{A_{ii}}^W(a) \sim e^{\frac{N}{2}(-a + \alpha \log a)}, \tag{60}$$

which could be generalized to the complex case as in [29]. For the off-diagonal matrix elements we derive the following probability distribution:

$$\log P_{A_{ij}}^W(a) \sim -\frac{N}{2} \left[ \alpha + \sqrt{4a^2 + \alpha^2} - \alpha \log\left( \frac{2a + \alpha + \sqrt{4a^2 + \alpha^2}}{2} \right) - \alpha \log\left( \frac{-2a + \alpha + \sqrt{4a^2 + \alpha^2}}{2} \right) \right]. \tag{61}$$

Let us finally note that the expression (12) in terms of the R-transform, which can be written as a series expansion starting from (11), establishes a relation between the cumulants of the random variable $A_{ii}$, the diagonal matrix element, or the cumulant of the off-diagonal matrix element $A_{ij}$ and the free cumulant $C_k$ of the matrix ensemble under consideration.

# 4 Conclusions

In this paper we study how an annealed measure 'deforms' the disorder, or, equivalently, how large deviations depend on rare realizations for the disorder. These realizations mimic the 'planted' ensemble, where the landscape is modified by creating an unusually deep valley, which is used in the theoretical study of inference problems, but in the annealed setting this valley is self-generated. This study was initially motivated by the observation that annealed computations are the most natural ones in spin-glass models that arise when one is interested in the probability of matrix elements of a matrix generated with a matrix model Hamiltonian. We discuss these in detail in the second part of the work, where our results show that large deviations (the tails of the distribution) of matrix elements occur when one or more eigenvalues have detached from the (typical) bulk of the spectrum of the matrix.

# Acknowledgements

We thank Davide Facoetti for useful discussions. This work is supported by "Investissements d'Avenir" LabEx PALM (ANR-10-LABX-0039-PALM) (EquiDystant project, L. Foini). J.K. is supported by the Simons Foundation Grant No 454943.

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
