# Peer review of "Annealed averages in spin and matrix models"

_SciPost Physics, doi:SciPost Phys. 12, 080 (2022)_

## Round 2 · Referee Report · Anonymous (Referee 1) · 2021-6-2

Strengths

see attached pdf report

Weaknesses

see attached pdf report

Report

see attached pdf report

Requested changes

see attached pdf report

Attachment

  • validity: high
  • significance: high
  • originality: -
  • clarity: high
  • formatting: excellent
  • grammar: excellent

Author:  Laura Foini  on 2021-10-26  [id 1881]

(in reply to Report 1 on 2021-06-02)
Category:
answer to question

We thank the referee for the appreciation to our work and the suggestions to improve the presentation.
In the attached file we reply to the comments and to the questions that the referee pointed out.
Kind regards,

Laura Foini

Attachment:

reply1_annealed.pdf

Anonymous on 2021-11-12  [id 1939]

(in reply to Laura Foini on 2021-10-26 [id 1881])

The authors have addressed most of the issues raised in my report. I would only like them to add one or more relevant references where the issue is discussed which they claim they cannot go into in detail here (the validity of the saddle point approach applied to integration over order(N) variables).

Author:  Laura Foini  on 2021-12-09  [id 2019]

(in reply to Anonymous Comment on 2021-11-12 [id 1939])

In standard random matrix ensembles the energy goes as N^2 times an action order one while the minimization is done over N eigenvalues. This implies that the large parameter in the action is much larger than the number of variables over which the saddle point is performed.
In the revised version we have written explicitly the factor N^2 in Eq. 2 and subsequent.
A complementary approach is to transform the integral over the discrete sets of eigenvalues as a functional integral over the density of eigenvalues and the saddle point is taken over such measure.
A detailed discussion of the terms at different order in N which appear in this procedure (and the fact that the entropic term order N is subleading) is discussed in Livan, Vivo, Novaes, Introduction to random matrices: theory and practice, Springer (2018), that we already cite in our work.
Mathematicians have made rigorous these results at least for a certain class of potentials, an example is discussed in the book E.B. Saff and V. Totik, Logarithmic Potentials with External Field, Springer-Verlag, 1997. We hope that this will reply to the concerns of the referee.

---

## Round 2 · Referee Report · Anonymous (Referee 2) · 2021-6-26

Strengths

1- Interesting phenomenology is found that is useful both for RMT specialists and statistical physicists.
2- The results are interesting and have a nice phenomenology.
3-I also think that the paper opens up quite a few perspectives for future study.

Weaknesses

1- The paper has a tendency to vagueness at some points and the overall presentation could be improved.
2-Exotic choice of notation

Report

In this paper the authors study an annealed version of the spherical SK model and show how it can be used to establish results on the statistics of the elements and sub matrices of random matrices. The paper is interesting and fairly well written and I found it very interesting although a bit difficult to follow at some of the exposition. There are points where the paper changes direction quickly, and the precise model under discussion could be better emphasized in some passages.
I think that the discussion of the phenomenon of eigenvalue detachment is a very interesting and intriguing phenomenon and could be related to some recent results in random matrix theory (RMT). The paper is suitable for publication after some improvements are made mainly at the level of presentation.

Requested changes

1-In the introduction the use of the dagger to note the dual vector to s is misleading as it suggest that the spin vector is complex.

2-The notation P(A_ij) is misleading it should be P_ij(a) as the probability distribution depends on the the pair ij.

3-The notation Ds is a bit odd - why not simply d{\bf s} as the integrals are finite dimensional and D is often used to functional integrals.

4-The phase transition in the spherical SK model does not always occur, it does for the Wigner semi-circle law but for density of states which do not vanish at the edge of the support of the density of states there should be no transition. It is basically the same transition as Bose Einstein condensation. I wonder what happens to the phenomenology of eigenvalue detachment in this case, does the whole scenario remain the same ?

5-In the annealed case the discussion of the formation of a molecule seems to be closely related to Baik-Ben Arous-Peche (BBP) transition for spiked random matrices J. Baik, G. Ben Arous, and S. Peche, Phase transition of the largest eigenvalue for nonnull complex sample covariance matrices, Ann. Probab. 33, 1643 (2005), this describes how a rank one perturbation to a random matrix can shift the largest eigenvalue. It might be work discussing this given the papers overall link with RMT.

6-I found the section 2.8 a bit out of place and expected 2.9 to be a continuation of this discussion, it would be better to put this in the introduction or the conclusion and it doesn’t really merit its own section.

7-The model 35 for p=2 is the standard SK model no ? It might be helpful to say this. In figure 6 it would be useful to recall that it is p=2 Ising. In general it might be useful to be more precise about which p=2 is being discussed.

8-Before equation 49 it would be clearer to call 49 the generating function of the probability distribution in equation 46.

The statement at the bottom of page 16 “Let us emphasize that this is a large N result, valid for r much smaller than N” is a bit vague. Should r be of order 1 or does N^1/2 work ?

  • validity: high
  • significance: high
  • originality: high
  • clarity: ok
  • formatting: reasonable
  • grammar: good

Author:  Laura Foini  on 2021-10-26  [id 1882]

(in reply to Report 2 on 2021-06-26)

We thank the referee for the appreciation to our work and the suggestions to improve the presentation.
Please find attached a pdf with the reply to the comments and the questions raised by the referee.
Kind regards,

Laura Foini

Attachment:

reply2_annealed.pdf

---

## Round 3 · Referee Report · Anonymous · 2021-10-26

Report
The authors have replied to the points made by myself and the other referee thoroughly and the revised version is suitable for publication.
Anonymous on 2021-11-29 [id 1984]
I am satisfied with the authors' response to my previous report. I would only like them to add at least one reference to papers where the conditions for validity of steepest descent in integrals where the dimensionality scales proportional to the large parameter in the exponent are discussed. The authors agree in their reply that this point is nontrivial and refer to literature on that topic, but without given pointers for the reader to that literature.

---

## Round 3 · List of Changes

In the revised version we have improved the presentation of our work following the remarks of the referees. A detailed list of changes can be found in the letters in reply to the referees.

You are currently on this page

---

## Round 4 · List of Changes

We have made a small modification to the text in order to reply to the referee question (see the reply to the comment)

---

## Editorial Decision

published